# Real-World Data on Second-Line Therapy with Ramucirumab for Metastatic Gastric Cancer: A Two-Center Study on Romanian Population

**DOI:** 10.3390/life13122300

**Published:** 2023-12-05

**Authors:** Diana Galos, Loredana Balacescu, Radu Vidra, Daniel Sur

**Affiliations:** 1Department of Medical Oncology, The Oncology Institute Prof. Dr. Ion Chiricuţă, 400015 Cluj-Napoca, Romania; dr.geni@yahoo.co.uk; 2Department of Genetics, Genomics and Experimental Pathology, The Oncology Institute Prof. Dr. Ion Chiricuţă, 400015 Cluj-Napoca, Romania; loredana_balacescu@yahoo.com; 3Postgraduate Program for Bio-Behavioral Integrative Medicine, Babes-Bolyai University, 400084 Cluj-Napoca, Romania; 4Department of Medical Oncology, Regional Institute of Gastroenterology and Hepatology Prof. Dr. Octavian Fodor, 400162 Cluj-Napoca, Romania; 5Department of Medical Oncology, University of Medicine and Pharmacy “Iuliu Hațieganu”, 400012 Cluj-Napoca, Romania

**Keywords:** gastric cancer, gastroesophageal junction cancer, metastatic disease, ramucirumab, paclitaxel, second-line treatment

## Abstract

(1) Background: Following the results of RAINBOW and REGARD trials, ramucirumab was approved as the standard second-line treatment for patients with advanced or metastatic gastric or gastroesophageal junction (GEJ) cancer, alone or in combination with paclitaxel. The present study aimed to evaluate the efficacy and safety of ramucirumab in the Romanian population during every-day clinical practice. (2) Methods: A two-center, retrospective, observational study evaluated patients with metastatic gastric and GEJ cancer treated with ramucirumab monotherapy or associated with paclitaxel. The patients were treated between 2018 and 2022 in two Romanian centers as follows: 18 patients underwent treatment with ramucirumab monotherapy, while 51 received the combined treatment regimen. Study endpoints included median progression-free survival (PFS), median overall survival (OS), and the evaluation of treatment-induced adverse events (AEs). (3) Results: In the study cohort (n = 69), the most frequent treatment-induced AE in the ramucirumab plus paclitaxel arm was hematological toxicity; the most common AE for patients treated with ramucirumab monotherapy was fatigue and headache. Overall, the median PFS was 4.7 months (95% CI: 3.4–5.9 months) and median OS was 18.23 months (95% CI: 15.6–20.7 months). PFS was correlated with the number of treatment cycle administrations, Eastern Cooperative Oncology Group performance status at treatment initiation, and metastatic site (visceral vs. peritoneal). OS was correlated with the number of treatment cycles administered and human epidermal growth factor *receptor*-2 status. (4) Conclusions: The results support the previously described toxicity profile for ramucirumab monotherapy or associated with paclitaxel and demonstrated a relatively superior median PFS.

## 1. Introduction

In the past few decades, the number of gastric cancer cases diagnosed in economically developed countries has experienced a significant decline [1]. However, gastric cancer remains an important health issue, representing the third leading cause of cancer-associated deaths worldwide [2]. 

According to Globocan database, in the year 2020, gastric cancer represented the fifth most prevalent cancer diagnosed in the male population of Romania. Local studies have also confirmed a boost in the number of male patients diagnosed with gastric cancer in recent years [3]. The reason behind the increase in gastric cancer cases could be tied to behavioral factors as well as environmental determinants. However, one of the most prominent risk factors associated with the surge in gastric cancer cases in the Romanian male population could be alcohol ingestion [3], with Romania being evaluated by WHO as one of the countries with the highest alcohol consumption in the world [4]. As far as environmental factors are concerned, the prevalence of *Helicobacter pylori* infections, despite experiencing a decline in recent years, remains significantly high in Romania [5]. In addition, Romanian environment exposes people to higher concentrations of particular carcinogens [6], of which some were associated with an increased risk of developing gastric cancer [7]. 

When diagnosed in an early stage (T1N0M0), gastric cancer is potentially curable through surgical or endoscopic resection [8]. Patients with advanced disease (>stage IA) require a multimodal approach, consisting of peri-operative chemotherapy [9]. Following the results from a German phase II–III study, the standard of care for stage IB-III disease dictates that four cycles of triple cytotoxic drug regimen (5-fluorouracil, leucovorin, oxaliplatin and docetaxel) should be administered before and after surgery [10]. Patients undergoing up-front surgery should receive adjuvant doublet chemotherapy consisting of oxaliplatin or docetaxel associated with fluoropyrimidines for a duration of 6 months [11]. The role of radiation therapy in the setting of gastric cancer remains controversial [12]. 

Despite recent progress surrounding therapies for metastatic disease, survival outcomes of patients diagnosed with metastatic gastric cancer remain unsatisfactory. Studies have confirmed a survival benefit following a first-line fluoropyrimidine-based chemotherapy associated with platinum compounds, while taxanes and irinotecan may also be considered as options [13]. Patients with HER2 (human epidermal growth factor *receptor*-2) overexpressing metastatic tumors benefit from the addition of trastuzumab as HER2-targeted treatment to standard chemotherapy [14]. Additionally, nivolumab administered with a doublet chemotherapy regimen was shown to improve overall survival (OS) in patients with programmed death-ligand 1 (PD-L1) overexpression and metastatic disease [15]. 

Following progression on first-line therapy, only 20–40% of patients continue treatment with second-line regimens [16]. In this setting, treatment options include paclitaxel, docetaxel and irinotecan, if not previously administered [17]. However, based on the results of two randomized, double-blind, placebo-controlled phase III trials, in 2014, the Food and Drug Administration (FDA) [18] and the European Medicines Agency (EMA) [19] approved ramucirumab as the standard of care for the second-line treatment of advanced gastric cancer and adenocarcinoma of the gastroesophageal junction (GEJ). Ramucirumab, a human IgG1 monoclonal antibody, is an angiogenesis inhibitor that selectively targets the vascular endothelial growth factor receptor (VEGFR)-2 [20]. Results from the phase III RAINBOW study have confirmed a significant increase in overall response rates (ORR) and progression-free survival (PFS) in patients receiving the combined treatment when compared with paclitaxel monotherapy [20]. In addition, OS increased significantly, from a median OS of 7.4 months in the paclitaxel arm to 9.8 months in the ramucirumab plus paclitaxel arm [20]. The addition of ramucirumab to paclitaxel also proved to delay symptom aggravation and functional status decay [21]. Ramucirumab monotherapy was studied in the phase III REGARD trial, demonstrating an improved OS compared with placebo (5.2 months vs. 3.8 months) [22]. 

In July 2018, the National Oncology Program of Romania covered ramucirumab for patients with metastatic gastric cancer who had experienced disease progression after previous chemotherapy with platinum-based compounds and/or fluoropyrimidine. The clinical experience of the current investigators with ramucirumab in this predefined indication has been somewhat conflicting, observing patients with encouraging survival outcomes, as well as patients with less favorable results following treatment. Since the clinical outcomes experienced in real-life setting seemed contrasting to those presented by clinical trials, the investigators proposed an evaluation of real-world data in order to better describe patients’ survival benefits following treatment with ramucirumab monotherapy or associated with paclitaxel. The research was designed as a retrospective, real-life investigation on a limited-sized cohort of Romanian patients. To our knowledge, this is the first study evaluating real-world experience with ramucirumab in metastatic gastric cancer in Eastern Europe and Romania. The current study included a population of patients with seemingly different risk factors than those included in previous studies conducted in other European regions. 

## 2. Materials and Methods

### 2.1. Study Design

Our team conducted a non-interventional, retrospective, two-center analysis of patients with metastatic gastric or GEJ cancer who were treated with ramucirumab monotherapy or associated with paclitaxel. The treatment was administered between January 2018 and December 2022 at The Oncology Institute Prof. Dr. Ion Chiricuță and The Regional Institute of Gastroenterology and Hepatology Prof. Dr. Octavian Fodor in Cluj-Napoca, Romania. The present research was verified in compliance with the principles of the Declaration of Helsinki, with all the participants providing written, informed consent. 

### 2.2. Patients

The research was designed to evaluate patients with a minimum age of 18 years, with the histological confirmation of gastric or GEJ carcinoma and metastatic disease confirmed through imaging studies. Most patients presented metastatic disease at diagnosis, and a subset of patients developed metastasis after primary treatment for early-stage disease. Patients were required to have undergone at least one previous line of chemotherapy before the initiation of ramucirumab monotherapy or in association with paclitaxel. All patients adhering to these criteria and receiving treatment with ramucirumab in the period between January 2018 and December 2022 were included in the study. Given the reputation of Cluj-Napoca as a high-performance medical center in Romania, the patients treated in the two institutes selected for the analysis were originally from various regions of the country, as Romanian patients have the possibility to choose the center where they prefer to receive treatment. Therefore, the cohort included in the study offers an accurate insight on the population of patients diagnosed with and treated for metastatic gastric cancer in Romania. 

Patients’ medical records were evaluated in order to collect baseline information: demographic data, Eastern Cooperative Oncology Group performance status (ECOG-PS), disease particularities (including HER2 mutation status), treatment information (surgical intervention for primary tumor, the administration of radiotherapy with curative intent, and prior systemic chemotherapy regimens), design of ramucirumab/ramucirumab plus paclitaxel administration (number of cycles and treatment de-escalation), treatment-induced toxicities, disease response, the date of progression and the date of decease (if applicable). 

### 2.3. Treatment

Patients underwent treatment with single agent, ramucirumab 8 mg/kg, on day 1, 15 or combination therapy with ramucirumab 8 mg/kg on day 1 and 15 and paclitaxel 80 mg/mp on day 1, 8 and 15. The treatment was administered intravenously, in a 28-day cycle, and continued until disease progression, decease, unacceptable toxicity or patients’ consent withdrawal. Over the course of treatment administration, the treatment suffered de-escalation when imperative, due to toxicity and decline in patient’s performance status.

### 2.4. Outcomes

Study endpoints included median PFS and median OS. PFS was characterized by the time interval elapsed between the initiation of ramucirumab/ramucirumab plus paclitaxel treatment and the first recorded disease progression or death of any cause. Disease progression was generally confirmed following imaging investigations, notably computed tomography, as well as magnetic resonance imaging and ultrasound. A subset of patients was confirmed to have experienced disease progression after being clinically evaluated and diagnosed with ascites or other clinical conditions indicative of disease evolution. OS was defined as the time period between the date of diagnosis of gastric and GEJ cancer and the date of decease. In addition, the research aimed to evaluate treatment-induced toxicities, evaluated according to The National Cancer Institute Common Terminology Criteria for Adverse Events (NCI CTACE), version 4.0. 

### 2.5. Statistical Analysis

The available data were collected in an Excel worksheet, and GraphPad Prism v. 8.0.2 (GraphPad Software, San Diego, CA, USA) was used for descriptive statistics and heatmaps. Survival data were analyzed in SPSS for Windows, v. 16.0 (SPSS Inc., Chicago, IL, USA). The survival curves were estimated using the Kaplan–Meier method, and survival distributions were compared with log-rank test. The effects of the main clinical and pathological variables on OS and PFS were investigated with Cox regression.

## 3. Results

The study analyzed data collected from 69 patients treated with ramucirumab monotherapy or in combination with paclitaxel. Patient and disease characteristics are summarized in Table 1. The study included 46 male and 23 female patients, with a median age at diagnosis of gastric and GEJ cancer of 62 years (range: 33–84 years). The majority of patients (78.3%) were diagnosed after the age of 50. Regarding tumor localization, 65 patients presented gastric tumors, and 4 patients presented tumors of the GEJ, with 97.2% of the described cases presenting the histology of adenocarcinoma. The most frequent histological subtype identified was tubular adenocarcinoma (37.7%), followed by poorly cohesive adenocarcinoma (26.1%). HER2 mutation was tested in 50 of the patients included in the study, of which 10 presented HER2-positive tumors. A total of 9 patients (13%) underwent neoadjuvant chemotherapy, and in 17 cases (24.6%), the primary tumor was resected. Thirteen patients (18.8%) benefited from fluoropyrimidine-based adjuvant chemotherapy, and in six cases (8.7%), radiotherapy was administered with curative intent. Regarding organ metastasis, the liver and peritoneum were the most common sites of metastasis identified. A significant percentage of patients (13%) had metastasis involving the peritoneum, as well as a visceral organ, while 15.9% of patients had metastasis in multiple visceral organs (≥2, peritoneum not involved). A total of 60 patients underwent first-line chemotherapy in the metastatic setting, of which 66.7% experienced disease progression in less than 6 months. Most patients (96.6%) followed a first-line metastatic treatment with fluoropyrimidine-based regimens, and the 10 patients with HER2-positive tumors benefited from antiHER2-targeted therapy. A fluoropyrimidine–platinum doublet was the dominant choice (58.2%), whereas 20% of patients received a combination of three cytotoxic agents (fluoropyrimidine, platinum, and anthracycline/taxane). Ramucirumab treatment was initiated in 55 patients (79.7%) with an ECOG PS < 1, and 14 patients (20.3%) were evaluated with an ECOG PS ≥ 1. The bulk of study participants (73.9%) were administered the combination therapy of ramucirumab plus paclitaxel, and the rest 26.1% were administered ramucirumab monotherapy. The duration of treatment with ramucirumab/ramucirumab plus paclitaxel was less than 6 months in the majority of patients (89.9%). Most patients discontinued treatment due to disease progression (43.5%), decease (26.1%) or inclusion in a palliative care program (11.6%). A total of five patients (7.2%) required discontinuation due to unacceptable treatment-induced toxicity. Four patients (5.8%) were lost from evidence, while four other patients were still receiving treatment at the time of study completion. 

Treatment-induced toxicities (Figure 1) were mostly correlated with paclitaxel administration. Hematological adverse events (AEs) were frequently observed in the study population, with anemia being most prevalent, followed by leucopenia, neutropenia and thrombocytopenia. Other paclitaxel-induced side effects noted in our study population were infusion-related reactions, neurologic toxicity and gastrointestinal and hepatic toxicity. When AEs were considered with respect to CTCAE v 4.0 grading, 24 events of grades 3 and 4 were noted, expressed as anemia, leucopenia, neutropenia, thrombocytopenia, infusion-related reactions and neurologic and hepatic impairment. Regarding ramucirumab administration, the most commonly observed adverse event was hypertension, followed by fatigue and headache, hemorrhage, gastrointestinal and renal impairment. Grade 3 hemorrhage occurred in four cases, and no grade 4 toxicity was associated with ramucirumab administration.

The patients included in the study required a careful monitoring of treatment-induced side effects, with some cases warranting dose and schedule modifications during administration. The design of the administration of treatment cycles is represented in Figure 2. Of the patients included in the study, only 11 patients received the full dose cycles of ramucirumab plus paclitaxel. Eighteen patients received ramucirumab monotherapy during the entire course of treatment, while the rest of the patients required dosage and scheduling adjustments according to patients’ tolerance and toxicity experienced.

A median OS of 18.23 months (547 days, 95% CI: 470.3–623.6) and a median PFS of 4.7 months (141 days, 95% CI: 104.4–177.5) was calculated for all evaluable patients enrolled in the study (Figure 3 and Figure 4).

In the univariate analysis, the number of ramucirumab plus paclitaxel administrations was significantly associated with OS (Figure 5A, Table 2) and PFS (Figure 6A, Table 3). The patients who received less than five cycles of treatment had a shorter OS (HR = 3.54, 95% CI: 1.67–7.49, p = 0.001) and PFS (HR = 8.43, 95% CI: 3.47–20.46, *p* < 0.0001) than patients who received more than five treatment cycles. A significant decrease (*p* = 0.014) in median OS was observed for the 52 patients in the arm with no surgery (16.33 months or 490 days) compared to the 17 patients with resected tumors (25.26 months or 758 days) (Figure 5B, Table 2). The difference between median PFS for patients with no surgery (3.96 months or 119 days) and those who underwent surgery (5.83 months or 175 days) had a marginal *p*-value of 0.057 (Figure 6B, Table 3). The log-rank comparison of the survival distribution according to the HER2 mutation status showed an increased median OS of 22.5 months (675 days) for HER2+ arm compared to 15.9 months (476 days) for HER2-group at a significance level of 0.053 (Figure 5C, Table 2), but no differences for PFS (Figure 6C, Table 3). Similarly, no statistically significant difference was noted in median OS (Figure 5D) and median PFS (Figure 6D) of patients with an ECOG PS < 1 or ≥1 at initiation of ramucirumab treatment. When evaluating the survival of patients who followed ramucirumab monotherapy vs. combined therapy with ramucirumab plus paclitaxel, no significant difference was noted in median OS (Figure 5E) or median PFS (Figure 6E). 

The potential predictors of OS and PFS were further evaluated in multivariate Cox regression model. The absence of HER2 mutation (HR = 4.04, 95% CI: 1.43–11.38, *p*-value = 0.008) and a lower number of treatments administered (HR = 3.40, 95% CI: 1.32–8.79, *p*-value = 0.011) were independent poor prognostic factors for OS in the multivariate analysis (Table 2). Considering only the HER2 effect in the univariate model (HR = 2.26, 95% CI: 0.97–5.26, *p*-value = 0.059), the differences in OS of negative vs. positive HER2 had a marginal *p*-value which exceeded the significance threshold, but when adjusted for covariates, the results became significant (*p*-value = 0.008) and the effect was even stronger (HR = 4.04) (Table 2). 

Moreover, the lower number of ramucirumab/ramucirumab plus paclitaxel cycle administrations, ECOG PS ≥ 1 and non-peritoneal metastasis were found as independent predictors of poor PFS in the multivariate model (Table 3). Although in the univariate analysis, surgery had a marginal *p*-value (*p* = 0.057), the resection of tumor did not prove to be an independent prognostic factor of PFS in the multivariate model (Table 3). In contrast, even if no significant differences were observed for ECOG PS (HR = 0.57, 95% CI: 0.28–1.15, *p*-value = 0.118) and metastasis (HR = 1.20, 95% CI: 0.67–2.17, *p*-value = 0.538) when they were considered separately, they were found as independent predictors for PFS when adjusted for covariates (ECOG PS-HR = 0.31, 95% CI: 0.13–0.75, *p*-value = 0.009; metastasis-HR = 2.28, 95% CI: 1.01–5.17, *p*-value = 0.049) (Table 3). Comparing with the univariate model, the HR value for the patients with a lower number of cycle administrations exhibited an increased value from 8.43 to 14.65, indicating that after accounting for the confounding effects of other factors, the impact of the number of administrations on survival is even stronger (Table 3). 

## 4. Discussion

Studies evaluating real-life data about the safety and efficacy of therapeutic agents offer valuable insight into treatment options and subsequent clinical outcomes. The conclusions drawn from such studies aid health care providers to better manage each case and patient, with profiles differing from those portrayed in randomized clinical trials. Treatment with ramucirumab in the setting of gastric and GEJ cancer has not yet been evaluated in Eastern Europe. In this context, the current study aims to describe a subgroup of patients of Romanian origin who benefited from ramucirumab therapy and the resultant therapeutic outcomes.

Ramucirumab is a human IgG1 monoclonal antibody that inhibits angiogenesis by selectively targeting the vascular endothelial growth factor receptor (VEGFR)-2 [23]. Once administered intravenously, the molecule binds and blocks the activation of VEGF receptor 2 and additionally blocks the binding of VEGF receptor ligands VEGF-A, VEGF-C and VEGF-D [24]. The anti-tumor medication has been approved as the standard treatment for adult patients with advanced or metastatic gastric and GEJ cancer after disease progression with previous platinum and fluoropyrimidine-based chemotherapy regimens, either as a monotherapy or in combination with paclitaxel [25]. 

The safety and efficacy of ramucirumab in the setting of gastric and GEJ cancer has been evaluated in a series of real-life studies. In Europe, the results from the Italian study RAMoss and Spanish study RAMIS were consistent with those observed during previous randomized clinical trials [26,27]. In addition, a real-world study conducted in the United States of America and one research based on a Korean expanded access program reached similar conclusions [28,29]. The largest study to date evaluating real-life experience with ramucirumab plus paclitaxel was conducted in Korea as a nationwide investigation, confirming the safety and efficacy of this therapeutic approach [30].

The patients in the Romanian population presented comparable baseline characteristics to those of patients included in randomized clinical trials. Our study reported a median age at diagnosis of gastric and GEJ cancer of 62 years, while the median age at randomization reported by RAINBOW trial and REGARD trial was 61 years and 60 years, respectively, implying a seemingly older median age of Romanian patients receiving the investigated treatment. However, both trials concluded a significant benefit of ramucirumab plus paclitaxel or ramucirumab monotherapy administration, in populations of <65 years, as well as ≥65 years. Accounting for more than half of the sample, the percentage of male patients in the current study (66.7%) was similar to that represented in the randomized trials (RAINBOW–69% male population, REGARD–71% male population). These data are in accordance with the profile universally described, making male sex patients almost twice as likely to develop gastric cancer [31]. Of note, the current study included a significantly smaller number of patients with GEJ adenocarcinoma (5.8%), making an accurate characterization of this subgroup of population difficult. AntiHER2 therapy was administered in 14.5% of patients included in this study, a significantly higher number compared to that reported by RAINBOW trial (9%). The percentage of HER2-positive tumors reported remains significantly lower than the incidence described in ToGA trial; however, this proportion could be partially explained by tumor location heterogeneity between the studies [14]. A total of 75.4% of patients included presented unresected primary tumors, as compared to 63% of patients in the RAINBOW trial and 73% of patients in the REGARD trial.

Similar to randomized clinical trials, a significant proportion of the population included in the current study had poor prognostic factors including poorly cohesive histological subtype, disease progression within 6 months after the start of first-line treatment, ECOG PS ≥ 1 and peritoneal metastasis or multiple metastatic sites. Progressive disease represented the main cause for treatment discontinuation in the presented study, as well as in both clinical trials. However, the current study noted a higher percentage of patient deaths occurring during treatment (26.1%). 

The toxicity profile was one of the main endpoints assessed by the current research. In general, the administration of ramucirumab therapy was well tolerated, with only 7.2% of patients requiring treatment discontinuation due to unacceptable toxicity as opposed to the discontinuation rates of 12% and 10.6% noted for the RAINBOW trial and the REGARD trail, respectively. Ramucirumab-assigned toxicity appeared comparable to that reported by randomized clinical trials, with most reported adverse events being hypertension, fatigue and hemorrhagic episodes. Our study reported fewer cases of clinically significant gastrointestinal and renal toxicity, and no occurrence of gastrointestinal perforations and venous thromboembolic events. As previously described, severe hematological events, particularly neutropenia, were more frequently reported in patients receiving paclitaxel. Neuropathy was also commonly described, as well as infusion-related reactions, as adverse events frequently described in the context of paclitaxel administration [32]. These findings are consistent with the safety profiles previously described in the literature. 

Median PFS observed in the current study was 4.7 months (95% CI: 3.4–5.9 months), and it was calculated including both treatment arms (ramucirumab monotherapy and ramucirumab plus paclitaxel), proving to be higher than that previously reported (4.4 months for the RAINBOW trial and 2.1 months for the REGARD trial). After including the follow-up data, the median OS was calculated to be 18.23 months (95% CI: 15.6–20.7 months). However, our study defined OS as the time elapsed between the diagnosis of gastric and GEJ cancer and the date of death from any cause, therefore describing a general overall survival of patients diagnosed with this particular cancer localization. 

Following the analysis of potential survival predictors provided by the current research, the absence of HER2 mutation was interestingly identified as a poor prognostic factor for OS. However, the literature data regarding the prognostic significance of HER2 expression in gastric cancer patients prove to be conflicting. Results from a vast meta-analysis have confirmed that HER2 overexpression is associated with a poor prognosis [33], while other studies have failed to support a correlation between HER2 expression and patient outcome [34,35]. Additionally, an ECOG PS ≥ 1 was associated with poor PFS, similar to the results described in clinical trials, where a poor ECOG PS was indicative of a decreased OS. However, contrary to the evidence provided by randomized trials, our study recognized non-peritoneal metastasis as an independent poor prognostic factor for PFS, while previous data suggest that the presence of peritoneal metastasis correlates with a reduced survival [36,37]. 

Naturally, the longer the PFS and OS periods experienced by patients, the more treatment cycles were therefore administered. This aspect raises the question whether certain tumor characteristics make patients more likely to respond to anti-VEGF therapy. An interesting biomarker that could have added value to the current research is microsatellite instability (MSI) testing. Microsatellite instable gastric tumors are commonly associated with older age at diagnosis and female gender, with localization in the middle and lower gastric body [38]. In addition, MSI cancers of the stomach were also shown to associate with a less frequent involvement of lymph nodes and a lower susceptibility to invade the serous anatomical layers, overall presenting a better prognosis [39]. Moreover, MSI tumors demonstrated a favorable response to immune checkpoint inhibitors [40]. MSI status has also been evaluated as an indicative of chemosensitivity of gastric tumors [38]. However, given that at the time when this study was conducted, immunotherapy for gastric and GEJ cancer did not benefit from any funding from the national health authorities, MSI status was not one of the biomarkers commonly assessed, neither at diagnosis, nor at disease progression after other lines of therapy. For this reason, data regarding genomic instability could not be assessed. However, describing a correlation between MSI status and response to ramucirumab therapy could prove beneficial in clinical settings. 

The current research faces a number of limitations. Since this study was designed as a real-life investigation, it is limited by the fact that the analysis was conducted retrospectively, in a non-randomized manner, on a limited number of participants. In addition, the study did not include a control group, as the aim of this research was to evaluate the safety and efficacy of ramucirumab therapy in daily clinical practice. Furthermore, during data collection, the investigators were faced with gaps in the information provided, making this study less rigorous. However, regardless of these limitations, data from real-life clinical experiences are essential for consolidating results obtained during clinical trials and confirming treatment efficiency amongst subgroups of patients. 

## 5. Conclusions

Real-life studies offer valuable information on treatment efficacy in populations with characteristics that may differ from those of patients included in rigorous clinical trials. Following the analysis of the data provided during our research, our findings support the toxicity profile previously described for ramucirumab therapy as single agent or in association with paclitaxel and demonstrates a marginally superior median PFS and a median OS of approximately 18 months; however, given the definitions of OS presented in the study, the result should be interpreted accordingly. Taking into consideration the results obtained during this investigation, ramucirumab-based treatment remains a valuable option for Romanian patients with advanced or metastatic gastric cancer and GEJ adenocarcinoma in a second-line setting. In addition, our results align with the conclusions drawn from randomized clinical trials, as well as other real-life studies, acknowledging ramucirumab as a useful agent in the second-line treatment of gastric or GEJ cancer. Given that the real-life investigations previously conducted included subgroups with distinct baseline characteristics from those presented in the current research, it may be established that ramucirumab efficacy benefits a significantly large population. 

## Figures and Tables

**Figure 1 life-13-02300-f001:**
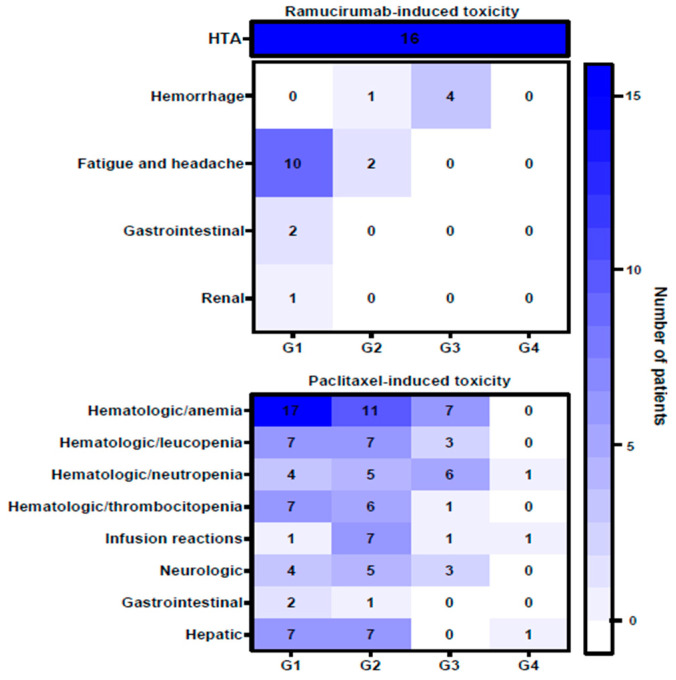
The heatmap of the treatment-induced toxicity. The grade of toxicity was ranked from G1 to G4. The intensity of the color is correlated with the number of patients who suffered from specified toxicity/grade (G1–G4).

**Figure 2 life-13-02300-f002:**
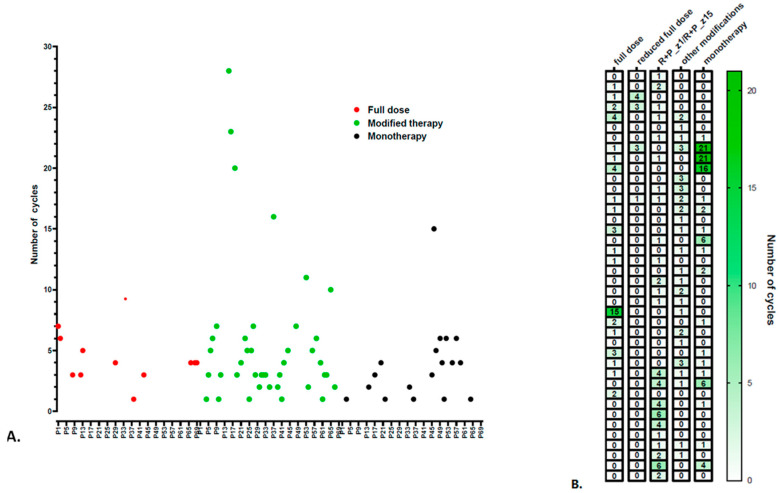
Design of ramucirumab cycles (**A**) Red dots represent the patients who received full dose (R + P Z1, P Z8, R + P Z15), green dots represent patients who received modified therapy, and black dots represent patients who received only ramucirumab. (**B**) Detailed modified therapy.

**Figure 3 life-13-02300-f003:**
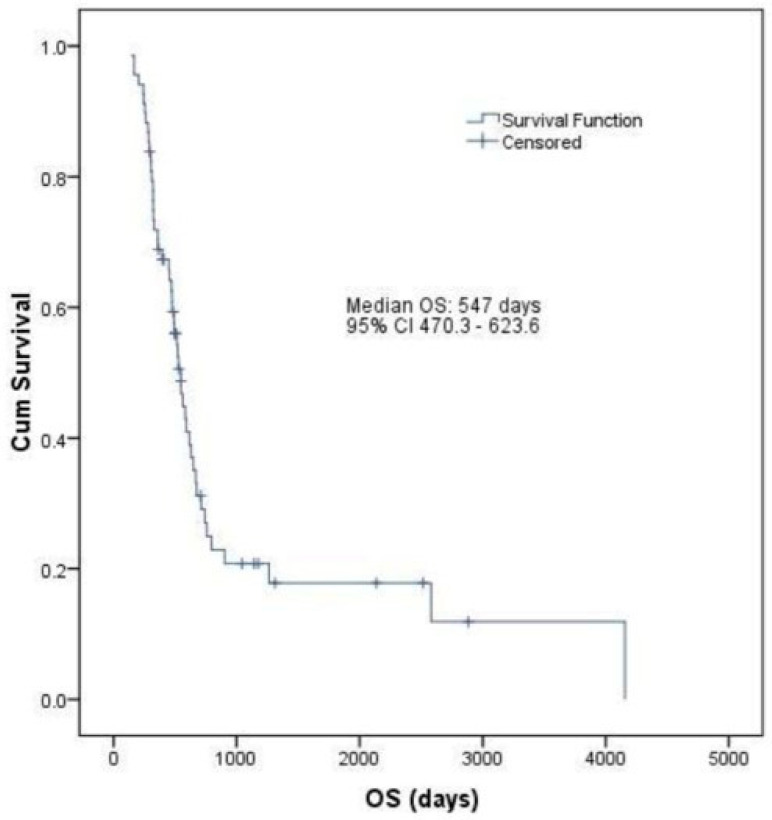
Kaplan–Meier curve of OS in all evaluable patients enrolled in the study (n = 69).

**Figure 4 life-13-02300-f004:**
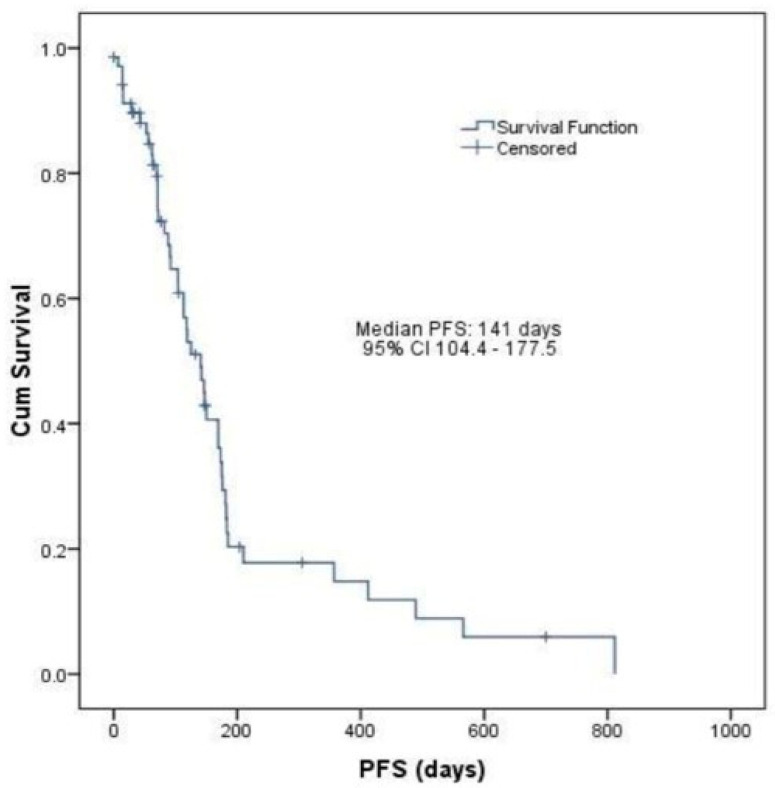
Kaplan–Meier curve of PFS in all evaluable patients enrolled in the study (n = 69).

**Figure 5 life-13-02300-f005:**
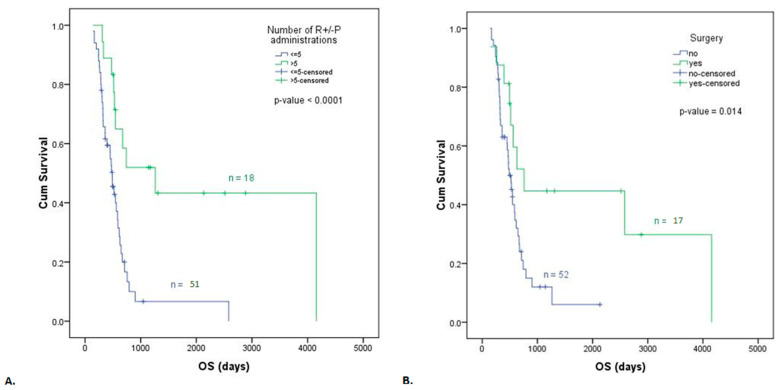
Kaplan–Meier curve for OS according to (**A**) the number of ramucirumab/ramucirumab plus paclitaxel administrations, (**B**) surgery, (**C**) HER2 mutation status, (**D**) ECOG PS, and (**E**) second-line metastatic ramucirumab/ramucirumab plus paclitaxel treatment.

**Figure 6 life-13-02300-f006:**
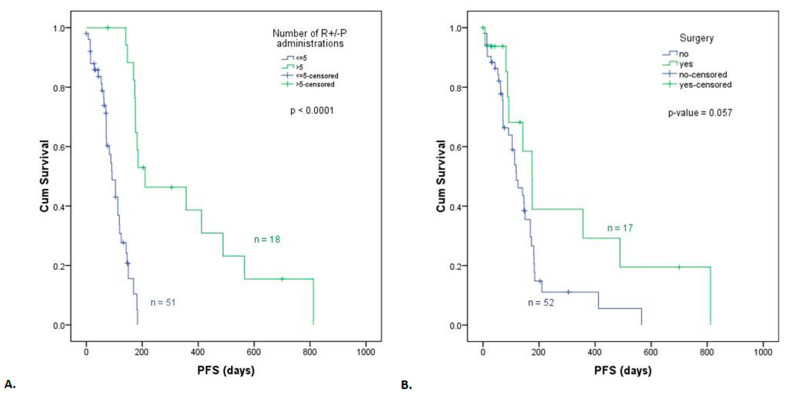
Kaplan–Meier curve for PFS according to (**A**) the number of ramucirumab/ramucirumab plus paclitaxel administrations, (**B**) surgery, (**C**) HER2 mutation status, (**D**) ECOG PS, and (**E**) second-line metastatic ramucirumab/ramucirumab plus paclitaxel treatment.

**Table 1 life-13-02300-t001:** Descriptive statistics of patients’ characteristics.

Variable	Number of Patients (n = 69)
** *Age* **	
*Median (range)*	62 (33–84)
<50	15 (21.7%)
≥50	54 (78.3%)
** *Sex* **	
F	23 (33.3%)
M	46 (66.7%)
** *Tumor localization* **	
Gastric	65 (94.2%)
GEJ	4 (5.8%)
** *Histology* **	
Adenocarcinoma	67 (97.2%)
Adenocarcinoma with neuroendocrine component	1 (1.4%)
Squamous	1 (1.4%)
** *Histologic subtype* **	
Mixed	4 (5.8%)
Mucinous	2 (2.9%)
Poorly cohesive	18 (26.1%)
Tubular	26 (37.7%)
NA	19 (27.5%)
** *HER2 mutation status* **	
HER2−	40 (58.0%)
HER2+	10 (14.5%)
NA	19 (27.5%)
** *Neoadjuvant treatment* **	
Yes	9 (13.0%)
No	60 (87.0%)
** *Primary tumor surgery* **	
Yes	17 (24.6%)
No	52 (75.4%)
** *Adjuvant treatment* **	
Yes	13 (18.8%)
No	56 (81.2%)
** *Radiation therapy with curative intent* **	
Yes	6 (8.7%)
No	63 (91.3%)
** *Site of metastasis* **	
Adnexa	2 (2.9%)
Bone	4 (5.8%)
Liver	19 (27.5%)
Lung	2 (2.9%)
Lymph nodes	5 (7.3%)
Peritoneum	16 (23.3%)
Peritoneum + other	9 (13.0%)
Skin	1 (1.4%)
Multiple sites (≥2, peritoneum not involved)	11 (15.9%)
	**Number of Patients (n = 13)**
** *Adjuvant treatment* **	
Fluoropyrimidine	1 (7.7%)
Fluoropyrimidine + platinum	7 (53.8%)
Fluoropyrimidine + platinum + anthracycline	2 (15.4%)
Fluoropyrimidine + platinum + taxane	3 (23.1%)
** *Time to progression on adjuvant treatment* **	
≤6 months	4 (30.8%)
>6 months	6 (46.1%)
NA	3 (23.1%)
	**Number of Patients (n = 60)**
** *First-line metastatic treatment* **	
Antimetabolite + platinum	1 (1.7%)
Platinum + taxane	1 (1.7%)
Fluoropyrimidine + platinum	35 (58.2%)
Fluoropyrimidine + platinum + anthracycline	6 (10.0%)
Fluoropyrimidine + platinum + taxane	6 (10.0%)
Fluoropyrimidine + topoisomerase I inhibitor	1 (1.7%)
Targeted therapy *	10 (16.7%) *
* Fluoropyrimidine + platinum + antiHER2	* 9 (15.0%)
* Fluoropyrimidine + platinum + anthracycline + antiHER2	* 1 (1.7%)
** *Time to progression on first-line treatment* **	
≤6 months	40 (66.7%)
>6 months	20 (33.3%)
	**Number of Patients (n = 69)**
** *ECOG PS at debut of R/R + P treatment* **	
<1	55 (79.7%)
≥1	14 (20.3%)
** *Second-line metastatic R/R + P treatment* **	
R	18 (26.1%)
R + P	51 (73.9%)
** *Duration of second-line metastatic R/R + P treatment* **	
≤6 months	62 (89.9%)
>6 months	7 (10.2%)
** *Reason for R/R + P treatment discontinuation* **	
Best supportive care	8 (11.6%)
Decease	18 (26.1%)
Disease progression	30 (43.5%)
Loss of evidence	4 (5.8%)
Toxicity	5 (7.2%)
During treatment	4 (5.8%)

GEJ: gastroesophageal junction; NA: not available; HER2−: human epidermal growth factor *receptor*-2 negative; HER2+: human epidermal growth factor *receptor*-2 positive; *: patients undergoing treatment with targeted therapy; ECOG PS: Eastern Cooperative Oncology Group performance status; R: ramucirumab; and R + P: ramucirumab + paclitaxel.

**Table 2 life-13-02300-t002:** Association of baseline characteristics with OS.

Variable	Kaplan–Meier Survival Analysis		Univariate Cox Regression Analysis		Multivariate Cox Regression Analysis	
	Median Survival Time (Days) (95% CI)	*p*-Value	HR (95% CI)	*p*-Value	HR (95% CI)	*p*-Value
HER2 mutation status (negative vs. positive)	476 (405.4–546.5) 675 (504.3–845.6)	0.053	2.26 (0.97–5.26)	0.059	4.04 (1.43–11.38)	0.008
Second-line R/R + P treatment (R vs. R + P)	547 (311.0–782.9) 529 (446.8–611.1)	0.708	0.89 (0.46–1.67)	0.709	0.82 (0.35–1.92)	0.654
Surgery (no vs. yes)	490 (412.7–567.2) 758 (414.8–1101.1)	0.014	2.57 (1.18–5.56)	0.017	2.31 (0.82–6.53)	0.114
Metastasis (non-peritoneal vs. peritoneal)	562 (477.5–646.5) 476 (377.1–574.9)	0.911	1.03 (0.57–1.87)	0.911	0.76 (0.37–1.57)	0.462
ECOG PS (<1 vs. >1)	584 (505.1–662.9) 394 (140.1–647.8)	0.084	0.57 (0.29–1.08)	0.088	0.53 (0.22–1.24)	0.140
Number of R/R + P administrations (≤5 vs. >5)	490 (417.0–562.9) 1263 (272.7–2253.2)	<0.0001	3.54 (1.67–7.49)	0.001	3.40 (1.32–8.79)	0.011

HER2: human epidermal growth factor *receptor*-2; R: ramucirumab; R + P: ramucirumab + paclitaxel; and ECOG PS: Eastern Cooperative Oncology Group performance status.

**Table 3 life-13-02300-t003:** Association of baseline characteristics with PFS.

Variable	Kaplan–Meier Survival Analysis		Univariate Cox Regression Analysis		Multivariate Cox Regression Analysis	
	Median Survival Time (Days) (95% CI)	*p*-Value	HR (95% CI)	*p*-Value	HR (95% CI)	*p*-Value
HER2 mutation status (negative vs. positive)	118 (94.0–141.9) 113 (56.8–169.1)	0.717	1.17 (0.50–2.69)	0.718	1.24 (0.51–3.00)	0.636
Second-line R/R + P treatment (R vs. R + P)	113 (0–226.4) 141 (107.7–174.2)	0.433	1.30 (0.67–2.52)	0.437	0.59 (0.22–1.61)	0.303
Surgery (no vs. yes)	119 (82.9–55.1) 175 (123.5–226.4)	0.057	2.04 (0.96–4.34)	0.063	1.92 (0.56–6.59)	0.298
Metastasis (non-peritoneal vs. peritoneal)	119 (72.9–165.1) 146 (102.5–189.4)	0.535	1.20 (0.67–2.17)	0.538	2.28 (1.01–5.17)	0.049
ECOG PS (<1 vs. >1)	147 (106.1–187.9) 91 (40.1–141.9)	0.112	0.57 (0.28–1.15)	0.118	0.31 (0.13–0.75)	0.009
Number of R/R + P administrations (≤5 vs. >5)	92 (67.7–116.2) 210 (1.8–418.1)	<0.0001	8.43 (3.47–20.46)	<0.0001	14.65 (3.67–58.56)	<0.0001

HER2: human epidermal growth factor *receptor*-2; R: ramucirumab; R + P: ramucirumab + paclitaxel; and ECOG PS: Eastern Cooperative Oncology Group performance status.

## Data Availability

Data are contained within the article.

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
