# Peer review of "Real-World Data on Second-Line Therapy with Ramucirumab for Metastatic Gastric Cancer: A Two-Center Study on Romanian Population"

_life, 2023, doi:10.3390/life13122300_

Round 1
Reviewer 1 Report
Comments and Suggestions for Authors
In general, the title of the article is good and the goal is expressive, but there is a flaw in the introduction and it needs to sequence the information gradually, and it also needs proofreading.
There is a flaw in selecting the patient sample, and the author says that the sample was taken between 2018 and 2022. How were the authors able to track the progress of treatment during these 5 years?
Dosages In the treatment section, the author did not refer to a reference documenting the amount of treatment and the details of its administration to the patient?
Comments on the Quality of English LanguageArticle need proofreading!
Reviewer 2 Report
Comments and Suggestions for Authors
Dear Author,
The manuscript is a well written, fair study, a useful addition in clinical oncology. As presented in introduction, ramucirumab proved to be already effective in various clinical trials, but it is relatively recently introduced in clinical therapy. This makes the “real-life” retrospective studies a useful addition. The study limitation, inherent to a retrospective study, were also very well highlighted. Overall, in my view the manuscript is worth to be published, the available data were properly analyzed, and the results are clinically relevant. In my opinion the manuscript does not need further revision.
Reviewer 3 Report
Comments and Suggestions for Authors
Manuscript Number: life-2697245
Comments to the Author
The paper entitled: “Real-world data on second-line therapy with Ramucirumab in metastatic gastric cancer: a two-center study on Romanian population” aimed to evaluate the efficacy and safety of ramucirumab (alone or combined with paclitaxel) in Romanian population during every-day clinical practice.
The manuscript is interesting, well-written, and has a potential for publication, however, some improvements/explanations need to be made before recommendation for acceptance.
In particular:
- The use of ramucirumab alone or combined with paclitaxel should be indicated also in the Title of the manuscript.
- The number (or percentage) of patients receiving ramucirumab as monotherapy or combined with paclitaxel should be specified in the Abstract (lines 21-22).
- Please specify ECOG and HER2 in extended form in the Abstract.
- The name “Helicobacter pylori” (Page 2, line 49) needs to be italicized.
- Abbreviations such as PD-L1 and OS (Page 2, Line 69) need to be written out in full when they are first mentioned in the Introduction section.
- Lines 104-105: The aim of this study, the type of study conducted (real-life investigation with several limitations), its novelty, and the type of treatment used (ramucirumab monotherapy or associated with paclitaxel) should be better described at the end of the Introduction section.
- Median progression-free survival and median overall survival are respectively abbreviated as PFS and OS in paragraph 2.4, while are indicated as mPFS and mOS in the Abstract. Please standardize.
- Several Figures are very difficult to read and need improvement, in particular, Figure 2A, Figure 5 and Figure 6 should be redrawn.
Reviewer 4 Report
Comments and Suggestions for Authors Dear authors,After reviewing the following manuscript entitled "Real-world data on second-line therapy with Ramucirumab in metastatic gastric cancer: a two-center study on Romanian population” (Life - 2697245), I sent the following comments and observations that the authors should attend to before its publication in this journal.
I appreciate the work of the authors, but please resolve the following data: The introduction is too long, I recommend the summary to a maximum of one page. Information from other studies on these pathologies and with the same or similar treatment should be brought to the Discussions. In the Conclusions, the importance of the results presented in this study should be specified more clearly. It was specified for patients from Romania but in general, which is relevant to the study.
